# The Performances of SNAQ, GLIM, mNICE, and ASPEN for Identification of Neurocritically Ill Patients at High Risk of Developing Refeeding Syndrome

**DOI:** 10.3390/nu14194032

**Published:** 2022-09-28

**Authors:** Na Liu, Xiao-Lin Zhao, Rui-Qi Xiong, Quan-Feng Chen, Yong-Ming Wu, Zhen-Zhou Lin, Sheng-Nan Wang, Tong Wu, Su-Yue Pan, Kai-Bin Huang

**Affiliations:** Department of Neurology, Nanfang Hospital, Southern Medical University, Guangzhou 510515, China

**Keywords:** refeeding syndrome, nutrition, RFS screening tools, neurocritical illness

## Abstract

We previously found that neurocritically ill patients are prone to refeeding syndrome (RFS), a potentially life-threatening complication. However, there is no unified or validated consensus on the screening tool for RFS so far. We aimed to validate and compare the performance of four screening tools for RFS in neurocritically ill patients. We conducted a single-center, observational, retrospective cohort study among neurocritically ill adult patients who were admitted to the neurocritical care unit (NCU), and who received enteral nutrition for 72 h or longer. They were scored on the Short Nutritional Assessment Questionnaire (SNAQ), the Global Leadership Initiative on Malnutrition (GLIM), the modified criteria of the Britain’s National Institute for Health and Care Excellence (mNICE), and ASPEN Consensus Recommendations for Refeeding Syndrome (ASPEN) scales to predict RFS risk via admission data. The performance of each scale in predicting RFS was evaluated. Logistic regression analysis was used to identify the independent risk factors for RFS, and they were added to the above scales to strengthen the identification of RFS. Of the 478 patients included, 84 (17.57%) developed RFS. The sensitivity of the SNAQ and GLIM was only 20.2% (12.6–30.7%), although they had excellent specificities of 84.8% (80.8–88.1%) and 86.0% (82.1–89.2%), respectively; mNICE predicted RFS with a sensitivity of 48.8% (37.8–59.9%) and a specificity of 65.0% (60.0–69.9%); ASPEN had the highest Youden index, with a sensitivity and specificity of 53.6% (42.4–64.4%) and 64.7% (59.8–69.4%), respectively. The Area Under the receiver operating characteristic Curves (AUC) of SNAQ, GLIM, mNICE, and ASPEN to predict RFS were 0.516 (0.470–0.561), 0.533 (0.487–0.579), 0.568 (0.522–0.613), and 0.597 (0.551–0.641), respectively. We identified age, Acute Physiology and Chronic Health Evaluation II (APACHE II), Sequential Organ Failure Assessment (SOFA), and Glasgow Coma Scale (GCS) score as independent risk factors of RFS, and the combination of GCS and age can improve the AUC of ASPEN to 0.664 (0.620–0.706) for predicting RFS. SNAQ, GLIM, mNICE, and ASPEN do not perform well in identifying neurocritically ill patients at high risk of RFS, although ASPEN appears to have relatively a good validity among them. Combining GCS and age with ASPEN slightly improves RFS recognition, but it still leaves a lot of room for improvement.

## 1. Introduction

Refeeding syndrome (RFS) refers to the biochemical and clinical symptoms, and metabolic abnormalities in malnourished patients undergoing refeeding, whether induced via oral, enteral, or parenteral feeding, and the hallmark of this phenomenon is hypophosphatemia [1,2,3]. RFS may lead to coma, lethal cardiac arrhythmias, heart arrest, acute respiratory failure, and even death [4,5]. Previous studies have connected a longer Intensive Care Unit (ICU)stay, increased mortality, and disability with the occurrence of RFS [6,7,8]. Although RFS is lethal, it can be prevented by identifying patients at high risk and restraining nutrition during refeeding to this group [9]. Therefore, there is an urgent need to develop a sensitive and specific screening tool to help to identify people who are at high risk of RFS.

Until now, the definition of RFS remains inconsistent, which has limited the development of screening tools for RFS, and the subsequent prevention and treatment of RFS. Nevertheless, based on some risk factors associated with RFS, such as a low body mass index, unintentional weight loss, and a history of alcohol, several screening scores have been developed to predict RFS. Among them, the Britain’s National Institute for Health and Care Excellence (NICE) [10] and the Short Nutritional Assessment Questionnaire (SNAQ) [11], are well-known. However, both the NICE and SNAQ scored poorly for sensitivity or specificity on retrospective validation analyses [12,13,14]. Friedli and colleagues [9] modified the NICE (mNICE) by adding additional criteria and identifying a very high-risk category. In 2020, the American Society for Parenteral and Enteral Nutrition (ASPEN) proposed consensus recommendations for the screening and management of patients who are at risk of developing or who have developed RFS, and suggested that other criteria set for diagnosing malnutrition such as the Global Leadership Initiative on Malnutrition (GLIM) may be predictive of RFS [15,16]. Unfortunately, the criteria of mNICE, ASPEN, and GLIM still lack wide validation in different populations, and have not been studied for their predictive value for RFS.

Neurocritical care units (NCUs) admit patients with life-threatening neurologic and neurosurgical disorders, or life-threatening neurologic manifestations of systemic diseases [17]. Neurocritically ill patients tend to be malnourished, owing to the characteristic of oropharyngeal dysphagia, impaired consciousness, perception deficits, cognitive dysfunction, and increased needs [18]. Enteral nutrition is the first choice for neurocritically ill patients, but it is more likely to develop RFS, due to the incretin effect of absorption of glucose [13]. In our previous study, we found a 17% incidence of RFS in neurocritically ill patients, and found that RFS was an independent risk factor for 6-month mortality [19]. Therefore, it is of great clinical significance to find an appropriate RFS screening tool for neurocritically ill patients.

In this study, we aimed to validate the performance of SNAQ, GLIM, mNICE, and ASPEN screening scales for identifying people who were at high risk of RFS in a cohort of neurocritically ill patients. In addition, we sought to identify the risk factors associated with the development of RFS and find out whether adding the risk factors to the above scales would improve their performance in recognizing risk patients of RFS.

## 2. Materials and Methods

### 2.1. Study Design and Population

This single-center, observational, retrospective cohort study was conducted at Nanfang Hospital, an affiliated teaching hospital of Southern Medical University in Guangzhou, China. This study was conducted in accordance with the Declaration of Helsinki, and the proposal was approved by the Nanfang Hospital’s Ethics Committee on clinical research of NFEC-2020-234, and the approval date was 16 October 2020. Informed consent was waived by the review board, due to its observational and retrospective properties, and all data were fully de-identified.

We screened all patients who were admitted to the NCU of Nanfang Hospital between January 2014 and September 2020. The general criteria for NCU admission were Glasgow Coma Scale (GCS) < 12, and/or admission for Acute Physiology and Chronic Health Evaluation (APACHE II) score > 15. Otherwise, patients were still admitted to NCU if they suffered from large hemisphere infarction, massive cerebellar infarction, severely intracerebral hemorrhage, or other life-threatening neurological diseases, as shown in our previous studies [19]. We only evaluated patients at the first admission if they were re-admitted during the same admission period. The body weight, height, energy intake, calculated daily targets, laboratory results, electrolytes, and glucose supplementation of all admitted patients could be accurately collected from our electronic medical record system. In addition, the nutritional data used to calculate the scores for the Nutrition Risk in Critically ill patients (NUTRIC) [20] and the malnutrition universal screening tool (MUST) [21] for nutritional and malnutritional risk assessment, respectively, were obtained within 24 h after NCU admission.

Patients were included in this study if they: (1) were on fully enteral feeding for > 72 h during the NCU stay; (2) had records of serum phosphates records before enteral feeding (baseline) and at 72 ± 12 h after feeding. Patients were excluded if they (1) had incomplete data on the nutritional provision or affecting the measurement of SNAQ, GLIM, mNICE, and ASPEN; (2) were aged > 85 or <18 years; (3) had serum phosphate <0.65 mmol/L (2.0 mg/dL) at NCU admission; (4) were lost to follow-up; (5) had end-stage malignant diseases; (6) had complications from diabetic ketoacidosis; or (7) had recent parathyroidectomy or were receiving renal replacement therapy, using phosphate binders, or undergoing therapeutic hypothermia. Patients finally included were divided into RFS and non-RFS groups according to the definition used in two recent high-quality studies [1,22].

### 2.2. RFS Risk Screening and Assessment

The SNAQ is a questionnaire regarding involuntary weight loss, appetite loss, and tube feeding or the recent use of supplemental drinks. The overall risk of malnutrition is established as follows: 0~1 = well-nourished, 2 = moderately malnourished, and 3~5 = severely malnourished [11]. The GLIM is a two-step model for malnutrition risk screening and assessment [16]. We quantified the GLIM criteria using a two-step for refeeding syndrome risk screening and assessment. The first step is to screen malnutrition risk by using MUST [21], and the second step is an assessment for diagnosis and grading the severity of malnutrition. Malnutrition risk is classified as no risk or low-to-moderate nutritional risk (MUST score < 2) and high nutritional risk (MUST score ≥ 2), which were assigned 0 and 1 point; malnutrition is recommended as stage 1 (moderate) and stage 2 (severe), which were assigned 2 and 3 points, respectively, in this study. The modified NICE criteria (mNICE) were an evidence-based and consensus-supported algorithm to identify inpatients at risk of RFS as being no risk (1 point), low risk (2 points), high risk (3 points), and very high risk (4 points) [9]. The ASPEN consensus criteria divided risks into moderate (1 point) and severe (2 points) [15]. The total scores of SNAQ, GLIM, mNICE, and ASPEN were calculated for each subject, based on electronic medical record data. All scorings were performed by an experienced resident to avoid variation between individuals (Appendix A).

### 2.3. RFS Definition and Data Collection

RFS was defined as the occurrence of new-onset hypophosphatemia within 72 h after starting nutritional support, which meant a drop of more than 0.16 mmol/L (0.5 mg/dL) for serum phosphate from any previous baseline and under the threshold of 0.65 mmol/L (2.0 mg/dL), consistent with that used in two recent well-conducted studies [1,22]. Baseline serum phosphate referred to phosphorus measured at the nearest time before starting nutritional support.

The electronic medical records of all candidates were reviewed rigorously, and demographic information, final diagnoses, and pre-existing comorbidities were recorded. We also collected the duration from onset to NCU admission, NCU stay, and hospital stay. The worst laboratory data within 24 h of NCU admission, including serum potassium, sodium, and magnesium, were recorded. Each patient had been assessed GCS at NCU admission, as well as APACHE II and SOFA scores. Two trained neurologists blinded to other study data obtained the information on mortality and functional recovery at 30 days and 6 months after NCU admission, through telephone review, where the poor outcome was defined as a modified Rankin Scale (mRS) > 3.

### 2.4. Nutrition Management

Every patient received a routine assessment of nutritional risk screening (NRS) 2002 at NCU admission [23,24]. Enteral nutritional support via nasogastric or nasointestinal (approximately 6%) tube was typically initiated within 24 h of admission to NCU, followed by a gradual climb to target calorie and protein levels at around 72 h. Meanwhile, we took measures to prevent high-risk patients from RFS, mainly by restricting calorie supply, in line with the NICE or mNICE guidelines [9,10]. Detailed information on the nutrition protocol is provided in Appendix A.

### 2.5. Statistical Analysis

Continuous variables were all expressed as median and interquartile range (IQR), due to a non-normal distribution. Categorical variables were presented as counts and percentages. The significant difference between two groups was compared using the Mann–Whitney U test, or in contingency tables using Pearson’s Chi-squared test and Fisher’s exact test (categorical variables). According to the cut-off value recommended by previous studies, patients in this study were considered to be at high risk of RFS when SNAQ score ≥ 2 [11], GLIM score ≥ 1 [16], mNICE score ≥ 3 [9], and ASPEN score ≥ 1 [15]. Then, the prognostic capabilities of the four screening scales were assessed by measuring their sensitivity, specificity, positive predictive value, and negative predictive value at the corresponding cut-off value. Receiver operating characteristic (ROC) curve analysis was performed to evaluate the discrimination of the screening tools, and the differences between the area under the curve (AUC) were examined using the DeLong test.

To explore potential predictors of RFS, univariate binary logistic regression analysis was first used to analyze the association between RFS and selected variables. Candidate variables that had a p-value of less than 0.05 were drawn into corresponding multivariable binary logistic regression models using the method of conditional forward, retaining in the final model only those that were specified prior, or that had an independent (*p* < 0.05) relationship with RFS. All statistical analyses were performed using SPSS version 25 (IBM, Armonk, NY, USA) and Medcalc version 15. For all analyses, *p* < 0.05 was considered as statistically significant.

## 3. Results

### 3.1. Patient Characteristics

We consecutively reviewed the medical records of 2053 patients admitted to the NCU of Nanfang hospital from January 2014 to September 2020, of which 545 met the inclusion criteria. After the exclusion of 67 patients, there were 478 in the final cohort (Figure 1). The overall study population consisted of patients with ischemic stroke (n = 192, 40.2%), intracerebral hemorrhage (n = 99, 20.7%), intracranial infection (n = 58, 12.1%), and other neurologic diseases (n = 129, 27.0%). The median (IQR) age was 59 (47–68) years, and 340 (71.1%) cases were male. The median (IQR) APACHE II score was 15 (11–19), and the median (IQR) length of the NCU stay was 8 (5–15). A total of 84 patients (17.6%) developed RFS within 72 h of admission to NCU.

### 3.2. Differences between the RFS and Non-RFS Groups

Baseline demographic and clinical characteristics, biochemical results, and outcomes of patients presenting with RFS and non-RFS were shown in Table 1. There were no differences in sex, body mass index (BMI), duration from the onset of diseases to our NCU admission, diabetes mellitus, mechanical ventilation at admission, MUST score, NRS 2002 score, the baseline level of serum potassium and sodium, caloric intakes on day 1 and day 3, and length of hospital stay between the RFS and non-RFS groups. It was remarkable that the RFS group tended to be older, have lower GCS scores, higher APACHE II, SOFA, and NUTRIC scores, have a lower baseline level of serum phosphate and magnesium, and have lesser caloric intakes on day 2, compared with those without RFS. Concerning the four RFS risk screening tools, only mNICE and ASPEN had a significant difference between these two groups. Unsurprisingly, the RFS group was associated with a higher requirement of in-hospital mechanical ventilation, longer NCU stay; and higher 30-day mortality, 6-month mortality, and likelihood of 6-month poor outcome (Table 1).

### 3.3. Validation and Comparison of Four Screening Tools for Predicting RFS

Patients were classified as being at high risk of RFS when they scored SNAQ ≥ 2, GLIM ≥ 2, mNICE ≥ 3, or ASPEN ≥ 1, and the corresponding sensitivity, specificity, and accuracy of the four screening tools were calculated (Table 2). For sensitivity, the ASPEN (53.6%) and the mNICE (48.8%) represented the highest and second highest values, respectively. The SNAQ and GLIM had excellent specificities of 84.8% and 86.0%, respectively, but they both had a low sensitivity of only 20.2%. The accuracy was 73% for SNAQ, 75% for GLIM, 62% for mNICE, and 63% for ASPEN. The Youden indexes were 0.050 for SNAQ score ≥ 2, 0.062 for GLIM score ≥ 1, 0.138 for mNICE score ≥3, and 0.183 for ASPEN score ≥ 1. Meanwhile, we calculated the positive predictive value, the negative predictive value, and the kappa value of all four tools on behalf of validating the authenticity (Table 2).

We further used ROC curve analysis to explore the discrimination of SNAQ, GLIM, mNICE, and ASPEN for predicting RFS (Figure 2). The results showed that all the screening tools had low AUCs (all < 0.6) in identifying RFS patients, and there was no significant difference between the AUCs of the four tools (Table 2 and Appendix A). Notably, the optimal cut-off values of GLIM, mNICE, and ASPEN identified from the ROC curves to distinguish high-risk RFS were the same as that selected in this study. Exceptionally, an optimal cut-off value of SNAQ was identified at ≥ 1 from the ROC curve, with a sensitivity of 46.4% (35.5–57.6%) and a specificity of 60.2% (55.1–65.0%).

### 3.4. Prediction Models of RFS

To explore the risk factors of RFS, relevant parameters were included into multiple multivariate models under the consideration of collinearity. In Model A, GCS, age, and ASPEN were found to be significantly associated with the development of RFS (Table 3). In other models (Appendix A), APACHE II and SOFA were found to be independently correlated with the occurrence of RFS, suggesting that disease severity is closely related to the occurrence of RFS in neurocritically ill patients. We then evaluated whether the combination of different disease severity scales and ASPEN, which had the largest AUC among the four screening tools, could improve the identification efficiency of RFS. We found that the combination of GCS (when ≤8, add 1 point) and age (when ≥65 years, add 1 point) with ASPEN (named mASPEN) did slightly, but significantly, improve the AUC to 0.664 (0.620–0.706) (Figure 2, Appendix A). Other combinations yielded similar results (Appendix A).

## 4. Discussion

This study further confirmed that RFS was not uncommon in neurocritically ill patients, with an incidence of 17.6%, which is consistent with our previous study [19]. Four screening tools for RFS, namely SNAQ, GLIM, mNICE, and ASPEN, did not perform well in identifying neurocritically ill patients at risk of RFS. Age and disease severity represented by GCS, APACHE II, and SOFA were independently associated with the occurrence of RFS, and the combination of disease severity and ASPEN criteria helped to improve RFS recognition, although the efficacy remained at acceptable levels.

The inconsistent definition and diagnosis of RFS still restricts the prevention and treatment of RFS, and related research on the condition. Given this, the American Society for Parenteral and Enteral Nutrition (ASPEN) Parenteral Nutrition Safety Committee and the Clinical Practice Committee proposed novel criteria for RFS diagnosis and stratification: a decrease in any one, two, or three out of serum phosphorus, potassium, and/or magnesium levels by 10–20% (mild), 20–30% (moderate), or >30%, and/or organ dysfunction resulting from a decrease in any of these, and/or due to thiamin deficiency (severe), occurring within 5 days of reintroduction of energy supplementation [16]. However, a recent study using the ASPEN criteria of RFS showed a 90% incidence of mild RFS, a 65% incidence of moderate RFS, and a 25% incidence of severe RFS among general hospitalized patients receiving enteral nutrition [25], indicating that the ASPEN criteria may be too broad. If the ASPEN criteria were used in our study population, the incidence of mild RFS could reach 100%. Therefore, we used the diagnostic criteria for RFS, consistent with those in two recent high-quality studies, namely, a drop of more than 0.16 mmol/L (0.5 mg/dL) in serum phosphate from any previous baseline, and under the threshold of 0.65 mmol/L (2.0 mg/dL) [1,22]. Consistent with our previous study [19], we found that the incidence of RFS in neurocritically ill patients was about 17%. It is worth mentioning that by using this definition of RFS, we excluded patients with baseline serum phosphate < 0.65 mmol/L (2.0 mg/dL), which might have some influence on the incidence of RFS and the performance of the four prediction tools in this study.

In validating the performance of screening tools for predicting RFS in neurocritically ill patients, we found that the sensitivity and positive predictive values of the SNAQ and GLIM were unsatisfactory, which may be attributed to the fact that these two screening or diagnosis tools are mainly used to identify populations that are at high risk of malnutrition or developed malnutrition without including other risk factors that closely contribute to RFS, such as alcoholism, drug use (insulin, chemotherapy, antacids, and diuretics), and decline in any electrolytes. In addition, in the original GLIM publication, there was no precise definition on how muscle mass and disease burden/inflammation were assessed, which provided uncertainty in explaining the results. Nevertheless, SNAQ and GLIM showed excellent specificity and negative predictive value. The work of Kraaijenbrink et al. [14] revealed similar findings in that the SNAQ had a low positive predictive value but a significant negative predictive value, although there were some differences between Kraaijenbrink’s and our study in terms of study population and the definition of RFS. This means that a low SNAQ score may facilitate the exclusion of RFS in daily practice, and encourage nutritionists and NCU physicians to provide adequate nutrition to low-risk populations promptly, potentially reducing the incidence of infection in NCU patients and mortality in the early phase of severe stroke [26]. As for GLIM, there has been no study to validate the predictive value of RFS, but a prospective study showed that the GLIM criteria performed excellently for diagnosing malnutrition, with a sensitivity of 86.6% and a specificity of 81.6% [27]. In general, SNAQ and GLIM may be more suitable for screening patients that are at risk of malnutrition than those at high risk of RFS, which the scales had been originally designed for.

In this study, mNICE and ASPEN performed with superior sensitivity and positive predictive values compared with SNAQ and GLIM, partly because the mNICE and ASPEN also included other risk factors that are known to associate with RFS, such as alcohol abuse or a medical history of long-term insulin use, and low baseline levels of potassium, phosphorus/phosphate, or magnesium before feeding [28,29]. RFS is a range of metabolic (most importantly, vitamin B1) and electrolyte (potassium, phosphorus/phosphate, or magnesium) alterations occurring as a result of the reintroduction and/or increased provision of calories after a period of decreased or absent caloric intake [15]. Alcohol abuse leads to the impaired absorption of vitamin B1, and low baseline potassium, phosphorus, and magnesium levels suggest that these electrolytes are not adequately stored in the body. These metabolic and electrolyte disorders will become more prominent when reintroducing nutrients or when increasing caloric supply. However, it is regrettable that mNICE and ASPEN only performed at medium predictive value. This may be due to the low proportion of patients with alcoholism in this study, and we excluded patients with baseline hypophosphatemia. Another possible interpretation may be that some useful biochemical markers were not included in these two screening tools, such as serum prealbumin, hemoglobin, IGF-1, leptin, and urea nitrogen [12,30,31]. In addition, the lack of specific criteria on how to assess subcutaneous fat and muscle mass loss presents a barrier to accurately scoring ASPEN. Nevertheless, mNICE and ASPEN scores differed significantly between the RFS and non-RFS groups, and the ASPEN showed the largest area under ROC for predicting RFS, indicating that the ASPEN criteria might be the most promising scale in identifying RFS risk patients, but further modification is required to improve the predictive efficacy.

In the present study, we found that low GCS, high APACHE II, and SOFA scores were associated with increased odds of RFS, indicating that RFS was also correlated with disease severity, which is consistent with our previous study [19]. The reason for this may be that the metabolic stress of critical illness leads to pathophysiological disturbance, a condition where metabolic disorders of electrolytes and vitamins are more likely to occur [1,3]. Ralib et al. [32] also showed that higher SOFA scores were associated with RFS, although no significant differences in APACHE II score were detected. Older age was an independent risk factor of RFS in this study, which has been repeatedly reported to be a risk factor for RFS [33]. Of note, we also found that the ability to predict RFS was significantly improved when these original screening scales such as ASPEN were combined with the severity of disease and age, although the improvements were still at an acceptable level. This suggests that some risk factors related to the occurrence of RFS have not been included in the existing screening tools, and it is necessary to further explore other factors that closely related to RFS to improve the early identification of the RFS high-risk population.

It should be noted that the day 2 calorie intake in the RFS group was significantly less than that in the non-RFS group, because we had restricted calorie intake for high-risk RFS patients according to our nutrition protocol (Appendix A) modified from the mNICE guidelines [9]. According to a recent high-quality randomized trial including 339 critically ill adults, conducted by Doig et al. [1], caloric restriction was an effective measure for preventing RFS in the intensive care unit. This trial is deemed the best quality evidence up to date regarding the treatment of patients with evidence of RFS [9]. However, the day 2 calorie intake (Kcal/kg) was not an independent risk factor in this study, possibly because a proportion of patients who developed RFS were not considered to be at high risk for RFS at admission, and did not restrict caloric intake.

There were several limitations to this study. Firstly, an inherent issue with the interpretation of this study is that it was a single-center retrospective study, and a larger number of potentially eligible participants could not be included, due to the missing serum phosphate data, either at NCU admission or 72 h after enteral nutrition support, which led to selection bias and residual confounding. Secondly, our new predictive models did not include some risk factors reflected in other studies [12,30,31], mainly because we did not routinely test these risk factors. Some items in the screening tools, such as loss of muscle mass, loss of subcutaneous fat, and detailed information on the previous nutrition support before admission to our NCU, were not accurately recorded in the electronic medical record, which may have led to an underestimation of the risk level of RFS in these patients, and so further validation through prospective studies is warranted. Thirdly, we have long recognized that RFS is not uncommon in neurocritically ill patients, and have routinely restricted calories in patients considered at risk of RFS, which may have influenced the actual incidence of RFS and the effectiveness of these screening tools. However, we found that it is still difficult to accurately identify high-risk patients with RFS, so it is still necessary to find a more objective, sensitive, and specific screening tool for identifying high-risk patients with RFS.

## 5. Conclusions

SNAQ, GLIM, mNICE, and ASPEN do not perform well in identifying neurocritically ill patients at high risk of RFS, although ASPEN appears to have relatively good validity among them. Combining GCS and age with ASPEN slightly improves RFS recognition, but still leaves a lot of room for improvement.

## Figures and Tables

**Figure 1 nutrients-14-04032-f001:**
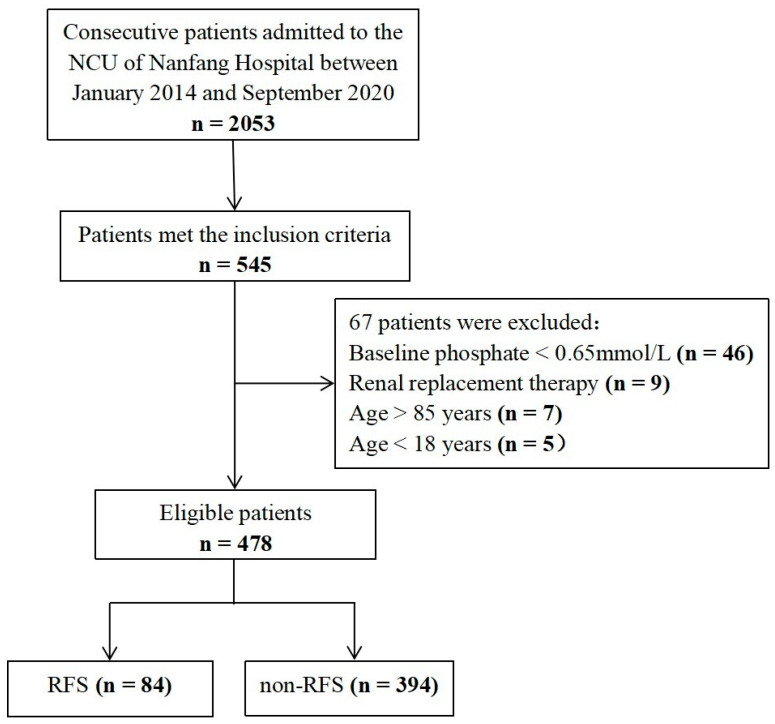
Patient inclusion flowchart.

**Figure 2 nutrients-14-04032-f002:**
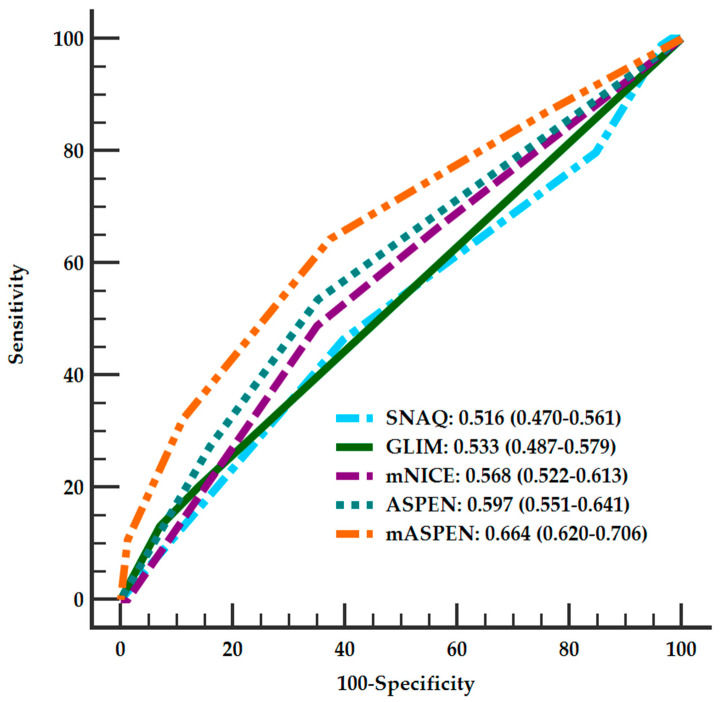
Receiver operating characteristic (ROC) curves of SNAQ, GLIM, mNICE, ASPEN, and mASPEN scores for the prediction of RFS. SNAQ, Short Nutritional Assessment Questionnaire; GLIM, Global Leadership Initiative on Malnutrition; mNICE, modified criteria of the Britain’s National Institute for Health and Care Excellence; ASPEN, American Society for Parenteral and Enteral Nutrition; mASPEN, the combination of GCS, age, and ASPEN; GCS, Glasgow Coma Scale; RFS, refeeding syndrome.

**Table 1 nutrients-14-04032-t001:** Patient characteristics, based on RFS and non-RFS groups.

Variables	RFS (n = 84)	Non-RFS (n = 394)	*p*
Age, years, median (IQR)	65 (51–73)	58 (47–67)	0.003
Female, n (%)	19 (22.9%)	119 (30.1%)	0.164
BMI, kg/m^2^, median (IQR)	22.9 (19.6–25.5)	23.2 (21.2–25.4)	0.433
Duration from the onset of diseases to our NCU, days, median (IQR)	2 (0–7)	3 (0–12)	0.181
History of alcoholism, n (%)	8 (9.5%)	59 (15.0%)	0.191
Hypertension, n (%)	52 (62.7%)	196 (49.6%)	0.023
Diabetes mellitus, n (%)	19 (22.9%)	77 (19.5%)	0.523
Heart disease, n (%)	16 (19.3%)	43 (10.9%)	0.040
Mechanical ventilation at admission, n (%)	10 (12.0%)	53 (13.4%)	0.737
APACHE II, median (IQR)	17 (13–22)	15 (11–19)	<0.001
GCS, median (IQR)	8 (6–11)	10 (7–12)	0.007
SOFA, median (IQR)	4 (3–7)	4 (2–6)	0.006
NUTRIC, median (IQR)	4 (3–5)	3 (2–4)	0.001
MUST, median (IQR)	2 (2–2)	2 (2–2)	0.107
NRS 2002, median (IQR)	4 (3–4)	4(3–4)	0.094
SNAQ, median (IQR)	1 (0–1)	1 (0–1)	0.631
GLIM, median (IQR)	1 (1–1)	1 (1–1)	0.1224
mMICE, median (IQR)	2 (1–3)	2 (1–3)	0.035
ASPEN, median (IQR)	1 (0–2)	0 (0–1)	0.001
**Baseline serum electrolytes**			
Phosphorous, mmol/L, median (IQR)	0.98 (0.86–1.14)	1.05 (0.92–1.21)	0.015
Potassium, mmol/L, median (IQR)	3.91 (3.53–4.24)	3.90 (3.63–4.20)	0.953
Sodium, mmol/L, median (IQR)	140 (135–144)	140 (137–143)	0.490
Magnesium, mmol/L, median (IQR)	0.84 (0.79–0.89)	0.86 (0.79–0.94)	0.039
** *Caloric intakes within the first 72 h* **			
Day 1, Kcal/kg, median (IQR)	8.22 (7.11–10)	8.45 (7.24–11)	0.21644
Day 2, Kcal/kg, median (IQR)	14.54 (11.93–18.11)	16.08 (12.85–20)	0.018
Day 3, Kcal/kg, median (IQR)	20.36 (15.71–25.08)	21.80 (16.97–25.8)	0.144
Mechanical ventilation in-hospital, n (%)	46 (55.4%)	163 (41.3%)	0.018
Length of hospital stay, days, median (IQR)	20 (11–33)	18 (12–29)	0.848
Length of NCU stay, days, median (IQR)	11 (6–18)	8 (5–14)	0.021
30-day mortality, n (%)	27 (32.5%)	66 (16.7%)	0.001
6-month mortality, n (%)	33 (39.8%)	77 (19.5%)	<0.001
6-month poor outcome (mRS > 3), n (%)	62 (73.8%)	207 (52.5%)	<0.001

RFS, refeeding syndrome; IQR, interquartile range; BMI, Body Mass Index; NCU, neurocritical care unit; APACHE II, Acute Physiology and Chronic Health Evaluation II; NUTRIC, Nutrition Risk in Critically ill patients; GCS, Glasgow Coma Scale; SOFA, Sequential Organ Failure Assessment; MUST, malnutrition universal screening tool; NRS 2002, nutritional risk screening 2002; SNAQ, the Short Nutritional Assessment Questionnaire; GLIM, the Global Leadership Initiative on Malnutrition; mNICE, modified criteria of the Britain’s National Institute for Health and Care Excellence; ASPEN, American Society for Parenteral and Enteral Nutrition; mRS, modified Rankin Scale. Baseline serum electrolytes, electrolytes measured at the nearest time before starting nutritional support.

**Table 2 nutrients-14-04032-t002:** The efficiency of the screening tools in identifying RFS.

	SNAQ (95%CI)	GLIM (95%CI)	mNICE (95%CI)	ASPEN (95%CI)	mASPEN (95%CI)
Sensitivity	20.2 (12.6–30.7)	20.2 (12.6–30.7)	48.8 (37.8–59.9)	53.6 (42.4–64.4)	63.1 (51.8–73.2)
Specificity	84.8 (80.8–88.1)	86.0 (82.1–89.2)	65 (60.0–69.6)	64.7 (59.8–69.4)	0.64 (58.9–68.7)
PPV	22.1 (13.7–33.2)	23.6 (14.7–35.3)	22.9 (17.1–29.9)	24.5 (18.6–31.4)	27.2 (21.2–34.1)
NPV	83.3 (79.2–86.7)	83.5 (79.4–87.0)	85.6 (81.0–89.3)	86.7 (82.2–90.3)	89 (76.8–92.3)
Accuracy	73	75	62	63	64
к	5.2	6.7	9.5	12.5	17.8
AUCs	0.516 (0.470–0.561)	0.533 (0.487–0.579)	0.568 (0.522–0.613)	0.597 (0.551–0.641)	0.664 * (0.620–0.706)

SNAQ, Short Nutritional Assessment Questionnaire; GLIM, Global Leadership Initiative on Malnutrition; mNICE, modified criteria of the Britain’s National Institute for Health and Care Excellence; ASPEN, American Society for Parenteral and Enteral Nutrition; mASPEN, the combination of GCS, age and ASPEN; GCS, Glasgow Coma Scale; PPV, positive predictive value; NPV, negative predictive value; CI, confidence interval; к, Kappa test statistic, percent of agreement; AUCs, area under the receiver operating characteristic curves. * *p* < 0.05 compared with SNAQ, GLIM, mNICE, and ASPEN.

**Table 3 nutrients-14-04032-t003:** Logistic regression analysis to identify predictors of RFS (Model A).

Parameter	Univariable Analysis		Multivariable Analysis	
	OR (95%CI)	*p*	OR (95%CI)	*p*
Hypertension	1.745 (1.074–2.834)	0.025	-	-
Heart disease	1.921 (1.023–3.606)	0.042	-	-
Day2 (Kcal/kg)	0.957 (0.919–0.996)	0.033	-	-
GCS	0.913 (0.852–0.979)	0.010	0.910 (0.848–0.978)	0.010
Age	1.025 (1.008–1.042)	0.004	1.027 (1.010–1.044)	0.002
ASPEN	1.573 (1.184–2.089)	0.002	1.597 (1.194–2.136)	0.002

ASPEN, American Society for Parenteral and Enteral Nutrition; GCS, Glasgow Coma Scale; Day2 (Kcal/kg), day 2 caloric intake; OR, odds ratio; CI, confidence interval.

## Data Availability

The data presented in this research are available upon request from the corresponding authors.

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
