# Peer review of "The Performances of SNAQ, GLIM, mNICE, and ASPEN for Identification of Neurocritically Ill Patients at High Risk of Developing Refeeding Syndrome"

_nutrients, 2022, doi:10.3390/nu14194032_

Round 1

Reviewer 1 Report

1. INTRO – well written, sets up research questions well.

2. METHODS

2.1 Study design: clear criteria for NCU admission. Data sources are stated. Clear inclusion and exclusion criteria including the length of enteral feeding and timing of phos draw. Please provide more detail on the refeeding protocol. This is critical for to support the scientific question since, in this case, the intervention (refeeding) is contributing to the outcome (RFS). Thus, please provide detail on whether enteral feeds were nasogastric (or other placement). Also provide density of the formula or liquid supplement (kcal/cc) and the rate of delivery (cc/hr). This information could be added to section 2.4. Please refer to Table 1 in this section, showing the starting caloric levels and advancement from Day 1 to 2 and 3. 

2.2 This text for this section is sufficient, however more detail is needed such that other investigators could recreate these scores and reproduce the findings. Could the authors present a table showing these four tools, with the actual items utilized for scoring, perhaps in supplementary materials? E.g. For ASPEN, were Mg and K evaluated or only Phos? (In section 2.3, I see Mg and Na mentioned.) 

2.3. Please clarify the definition of RFS. Was hypophosphatemia defined as a drop of > 0.16 mmol/L AND under the threshold of 0.65 mmol/L? It is not crystal clear whether a drop of >0.16 would be counted as RFS alone or whether any serum level < 0.65 would be counted as RFS alone. Please also include serum levels in mg/dL for the U.S. readership as well as mmol/L (unless that is not the convention for this journal).

2.4. As stated in 2.1, it is absolutely critical to provide more detail about the refeeding approach including caloric density of the formula and rate of delivery. Please refer to Table 1 in this section. Authors state here that they restricted calories in high-risk patients. Please provide details on exactly how these patients were deemed high risk (e.g. provide score thresholds or weight cut-offs). If this initial risk assessment was derived from MUST or NUTRIC, please say so. Then please provide details on the restricted calorie protocol, what was the restricted caloric level and slowed rate of advancement?  

2.5 Statistical analysis is well written; scoring rubric is clear and well referenced. I am not skilled with ROC analysis and therefore cannot comment on this approach. Forward approach to logistic regression is sound and significance level for variable retention is clearly stated.  

3. RESULTS 

3.1 Patients characteristics are clear and flow diagram is helpful. As stated above, need scientific reasoning/justification for excluding patients with phos < 0.65 mmol/L. 

3.2. Please comment on the significantly lower kcal on Day 2 in the RFS group. Does this reflect the ”restricted kcal protocol”? into the 

3.3. No comments.

3.4. No comments.

DISCUSSION 

Limitations are acknowledged. 

Author Response

Q1. INTRO – well written, sets up research questions well.

Response: We appreciate your kind comment.

Q2.1 Please provide more detail on the refeeding protocol. This is critical for to support the scientific question since, in this case, the intervention (refeeding) is contributing to the outcome (RFS). Thus, please provide detail on whether enteral feeds were nasogastric (or other placement). Also provide density of the formula or liquid supplement (kcal/cc) and the rate of delivery (cc/hr). This information could be added to section 2.4. Please refer to Table 1 in this section, showing the starting caloric levels and advancement from Day 1 to 2 and 3. 

Response: We thank the reviewer for this insightful comment. As per your suggestion, we have added our nutrition protocol in the supplementary materials (Supplementary Fig.1), which is mainly formulated according to the NICE and mNICE guidelines[1, 2], as shown in the following figure. According to our feeding protocol, the rate of initiation of enteral nutrition and the daily rate of increase were different for subjects with different levels of refeeding risk. For the selection of enteral nutrition formulas, we selected the diabetes-specific formulas (Enteral Nutritional Emulsion [TPF-D], FRESENIUS KABI, 0.9Kcal/cc) for diabetics and other formulas (JAVITY [TPF-FOS], Abbott, 1.0Kcal/cc; TPF, NUTRICIA, 1.0Kcal/cc) for non-diabetics.

A majority of patients in this study received enteral nutrition through a nasogastric tube, with early nasointestinal tube feeding (approximately 6% in this study) only in patients at very high risk of aspiration. We have added the enteral nutrition pathway (nasogastric tube or nasointestinal tube) to section 2.4.

Supplementary Fig. 1. The refeeding protocol for neurocritically ill patients.

Q2.2 This text for this section is sufficient, however more detail is needed such that other investigators could recreate these scores and reproduce the findings. Could the authors present a table showing these four tools, with the actual items utilized for scoring, perhaps in supplementary materials? E.g. For ASPEN, were Mg and K evaluated or only Phos? (In section 2.3, I see Mg and Na mentioned.)

Response: As per your kind suggestion, we have put the items of the four tools in the supplementary materials (Supplementary Table S1-S7). As for ASPEN, Mg and K were all evaluated. All items had been evaluated for all screening scores, except two items which were loss of subcutaneous fat and loss of muscle mass. The scores of loss of subcutaneous fat and loss of muscle mass were deemed as 0 for all subjects, which was a weakness of the study and has been addressed in the Limitation section of the manuscript.

Q2.3 Please clarify the definition of RFS. Was hypophosphatemia defined as a drop of > 0.16 mmol/L AND under the threshold of 0.65 mmol/L? It is not crystal clear whether a drop of >0.16 would be counted as RFS alone or whether any serum level < 0.65 would be counted as RFS alone. Please also include serum levels in mg/dL for the U.S. readership as well as mmol/L (unless that is not the convention for this journal).

Response: Yes, in this study, RFS was defined as a drop of > 0.16 mmol/L (0.5 mg/dL) and under the threshold of 0.65 mmol/L (2.0 mg/dL). It was consistent with that used in two recent well-conducted studies[3, 4]. To minimize the misunderstanding to the readers, we have changed the expression to “RFS was defined as the occurrence of new-onset hypophosphatemia within 72 hours after starting nutritional support, which meant a drop of more than 0.16 mmol/L (0.5 mg/dL) on serum phosphate from any previous baseline and under the threshold of 0.65 mmol/L (2.0 mg/dL)”. We also have described the serum levels in mg/dL in the revised manuscript.

Q2.4 As stated in 2.1, it is absolutely critical to provide more detail about the refeeding approach including caloric density of the formula and rate of delivery. Please refer to Table 1 in this section. Authors state here that they restricted calories in high-risk patients. Please provide details on exactly how these patients were deemed high risk (e.g. provide score thresholds or weight cut-offs). If this initial risk assessment was derived from MUST or NUTRIC, please say so. Then please provide details on the restricted calorie protocol, what was the restricted caloric level and slowed rate of advancement?

Response: As a response to Q 2.1, the details of our refeeding protocol are shown in Supplementary Fig.1. We used the mNICE guidelines for the initial RFS risk assessment [1]. MUST and NUTRIC scores were just performed for malnutrition and nutritional risk screening.

Q2.5 Statistical analysis is well written; scoring rubric is clear and well referenced. I am not skilled with ROC analysis and therefore cannot comment on this approach. Forward approach to logistic regression is sound and significance level for variable retention is clearly stated.  

Response: Thank you.

Q3.1 Patients characteristics are clear and flow diagram is helpful. As stated above, need scientific reasoning/justification for excluding patients with phos < 0.65 mmol/L. 

Response: There is no widely accepted, unified definition for refeeding syndrome. In this study, to observe the incidence of RFS and the prediction of RFS by the four tools, the definition of RFS was adopted from two recently published high-quality studies, namely, RFS was the decrease of serum phosphorus level from above 0.65 mmol/L to below 0.65 mmol/L after reintroduction of nutrients and the decrease level was more than 0.16 mmol/L. According to our definition, if patients with baseline phosphorus < 0.65 mmol/L were included, it was impossible to determine which patients would develop RFS during refeeding. We have included an explanation of this problem in the Discussion section of the revised manuscript (Line 286-289).

Q3.2 Please comment on the significantly lower kcal on Day 2 in the RFS group. Does this reflect the ”restricted kcal protocol”? into the

Response: The significantly lower kcal on Day 2 in the RFS group does result from the “restricted kcal protocol”, which reflects on the restricted kcal protocol shown in Supplementary Fig. 1. We also have addressed this issue in the 6th paragraph of the Discussion section.

References:

  1. UK, N.C.C.F. Nutrition Support for Adults: Oral Nutrition Support, Enteral Tube Feeding and Parenteral Nutrition; National Collaborating Centre for Acute Care (UK): London, 2006.
  2. Friedli, N.; Stanga, Z.; Culkin, A.; Crook, M.; Laviano, A.; Sobotka, L.; Kressig, R.W.; Kondrup, J.; Mueller, B.; Schuetz, P. Management and prevention of refeeding syndrome in medical inpatients: An evidence-based and consensus-supported algorithm. NUTRITION 2018, 47, 13-20.
  3. Doig, G.S.; Simpson, F.; Heighes, P.T.; Bellomo, R.; Chesher, D.; Caterson, I.D.; Reade, M.C.; Harrigan, P.W.J. Restricted versus continued standard caloric intake during the management of refeeding syndrome in critically ill adults: a randomised, parallel-group, multicentre, single-blind controlled trial. The Lancet Respiratory Medicine 2015, 3, 943-952.
  4. Olthof, L.E.; Koekkoek, W.; van Setten, C.; Kars, J.; van Blokland, D.; van Zanten, A. Impact of caloric intake in critically ill patients with, and without, refeeding syndrome: A retrospective study. CLIN NUTR 2018, 37, 1609-1617.

Reviewer 2 Report

The article summarizes the experience of physicians working in a neurocritical care unit (NCU) regarding malnutrition and development of refeeding syndrome.

There is a major criticism regarding the study design. The GLIM criteria are used as and compared with malnutrition screening tools. However, GLIM is a set of criteria that establishes the diagnosis of malnutrition, from mild to severe, not simply another screening tool. In fact, GLIM requires as a first step to do a nutritional screening with one of the approved screening tools. A patient may be at nutrition risk but not malnourished. The screening could be positive, but he/she does not fulfil the diagnostic criteria of malnutrition. Therefore, the authors should rewrite the article according to the correct  GLIM concept. For example, GLIM should be on a different category in Table 1. Or lines 211-214, or 291-2932: GLIM is not designed for screening.

Until the Statistical Analysis section (2.5) in line 144 there is not mention of the two groups that the authors have studied. These studies should be better described earlier. They should be better mentioned in Section 2.1

There are other issues that should be clarified:

1.      How many people were involved in the collection of data?. The manuscript says that all scoring was performed by an experienced resident, but the study took from January 2014 to September 2020 and it seems strange that the same person did the scoring of all patients.

2.      How the weight and weight loss was assessed in patients admitted to a NCU should be better explained. Many of these patients cannot stand. This information is essential for most screening tools, less so with SNAQ.

3.      NUTRIC and MUST appear in line 138, but probably they should be mentioned in line 95.

4.      The authors should better explain the decrease in the number from 2053 to 545 patients in the algorithm of Figure 1. How has the risk of bias been controlled when including/excluding patients? This is an essential information

5.      Line 183. Is there a reason why patients with RFS had a lower caloric intake than patients without RFS? It should appear in the discussion

6.      In Table 1 daily kcal should be better described as kcal/kg. Otherwise, it is rather meaningless.

7.      Line 198, consider changing “classified” for “identified”.

8.      Line 264, consider changing “calories” by “energy supply”, for example

9.      9. Lines 294-297, 315-316, need rewriting.

As the findings of the authors is that malnutrition screening tools are poor predictors of RFS, they could speculate in the discussion about the pathophysiology of RFS and nutrient provision, independently of the nutrition status.

Author Response

Response to Reviewer 2:

Q1. There is a major criticism regarding the study design. The GLIM criteria are used as and compared with malnutrition screening tools. However, GLIM is a set of criteria that establishes the diagnosis of malnutrition, from mild to severe, not simply another screening tool. In fact, GLIM requires as a first step to do a nutritional screening with one of the approved screening tools. A patient may be at nutrition risk but not malnourished. The screening could be positive, but he/she does not fulfil the diagnostic criteria of malnutrition. Therefore, the authors should rewrite the article according to the correct GLIM concept. For example, GLIM should be on a different category in Table 1. Or lines 211-214, or 291-2932: GLIM is not designed for screening.

Response: Thank you for the insightful suggestion. Yes, GLIM is a set of criteria that establishes the diagnosis of malnutrition, from mild to severe. However, another fact is that malnourished patients are a high-risk population for developing RFS. If we look at the items used by the GLIM, we could find that BMI, weight loss, and reduced muscle loss are also the indicators used by the RFS screening tools mNICE and ASPEN, which also indicates that GLIM has the potential to be used for RFS screening. The ASPEN Consensus Recommendations for Refeeding Syndrome has proposed that other criteria set for diagnosing malnutrition, such as GLIM, may be predictive of RFS[1]. A recent prospective study regarding RFS and COVID-19 disease conducted by Shariatpanahi et al.[2] revealed that of the 268 patients who were at risk of RFS as identified by the ASPEN criteria, 73% were malnourished in terms of GLIM. These findings prompted us to use GLIM for nutritional assessment to identify patients at risk of refeeding syndrome. We have revised the statement of why using GLIM for RFS risk screening in the Introduction section.

      As for the use of GLIM, we totally agree with the reviewer and follow the original GLIM diagnostic scheme for screening, assessment, diagnosis and grading of malnutrition[3]. In the first step, we used MUST to screen malnutrition risk and found that all the eligible participants at least scored 2 (as shown in the following Table1), that meant all participants were in high risk of malnutrition and were able to move to the second step, diagnosis assessment and diagnosis of malnutrition according to the phenotypic or etiologic criteria of GLIM. Because this study included patients with acute neurocritical illness, they all met at least one of the etiological criteria, e.g., disease burden /inflammatory condition. Thus, all the participants could move to the severity grading of malnutrition. Malnutrition was assessed as stage 1 (moderate) and stage 2 (severe) based on phenotypic criteria. For statistical convenience and direct comparison of the validities with the other tools, in this study, moderate and severe malnutrition were scored 1 and 2 points, respectively. These GLIM processing stage generated the data shown in the Materials and methods section (114-121). To reduce misunderstanding, we have revised the relevant description accordingly.

Table 1 The MUST score of all participants.

0 scores

1 score

2 scores

 3 scores

4 scores

5 scores

6 scores

Total

Frequency

0

0

403

43

25

3

4

478

Percent (%)

0

0

84.3

9.0

5.2

0.6

0.8

100

Q2. Until the Statistical Analysis section (2.5) in line 144 there is not mention of the two groups that the authors have studied. These studies should be better described earlier. They should be better mentioned in Section 2.1

Response: We have added the grouping information in the last part of Section 2.1.

Q3. How many people were involved in the collection of data? The manuscript says that all scoring was performed by an experienced resident, but the study took from January 2014 to September 2020 and it seems strange that the same person did the scoring of all patients.

Response: There were 4 persons involved in data collection. R.X., Q.C., and T.W. were responsible for screening subjects according to the inclusion and exclusion criteria and obtaining objective indicators such as demographic data and laboratory results. For the final 478 subjects included, the SNAQ, GLIM, mNICE, and ASPEN scores were evaluated by N.L. alone to achieve the consistency of RFS evaluation.

 Q4. How the weight and weight loss was assessed in patients admitted to a NCU should be better explained. Many of these patients cannot stand. This information is essential for most screening tools, less so with SNAQ.

Response: For patients who cannot stand, the weight on admission was measured by the weight bed. The weight change was obtained from the family members or the care provider, which had been detailed documented in our electronic medical record.

Q5. NUTRIC and MUST appear in line 138, but probably they should be mentioned in line 95.

Response: As suggested by the reviewer, we have revised the manuscript accordingly.

Q6. The authors should better explain the decrease in the number from 2053 to 545 patients in the algorithm of Figure 1. How has the risk of bias been controlled when including/excluding patients? This is an essential information

Response: Thanks for the constructive suggestion. A total of 1508 patients were not included during the initial screening stage, as shown in the figure below. Of them, 1192 (79%) did not receive fully enteral feeding or the enteral feeding time was less than 72 h during their NCU stay. They mainly consisted of patients with acute ischemic stroke who received endovascular treatment and required neurocritical care but recovered quickly and may not require enteral feeding longer than 72 h, as well as those with rapidly deteriorating hemodynamics or neurological function who cannot tolerate enteral nutrition. These patients either received neurocritical care for too short a time or their condition changes were mainly caused by primary disease or severe infection, so they may not suitable for the representative population of neurocritically ill patients receiving enteral nutrition to be observed for susceptibility to refeeding syndrome. Therefore, excluding such patients should not bring significant bias to the target population.

Other reasons that were ruled out in the initial screening were the absence of baseline serum phosphorus (n = 39) and 72 h follow-up serum phosphorus (n = 277). This might lead to information bias, which we have already discussed in the Limitation part of the manuscript.

Fig. 2. Patient inclusion flowchart.

Q7. Line 183. Is there a reason why patients with RFS had a lower caloric intake than patients without RFS? It should appear in the discussion

Response: The reason that the day 2 calorie intake in the RFS group was significantly less than that in the non-RFS group was that we had restricted calorie intake for high-risk RFS patients according to our nutrition protocol (Supplementary Fig. 1) modified from the mNICE guidelines. According to a recent high-quality randomized trial including 339 critically ill adults conducted by Doig et al[4]., caloric restriction was an effective measure to prevent RFS in the intensive care unit. This trial is deemed the best quality evidence up to date regarding the treatment of patients with evidence of RFS[5]. However, the day 2 calorie intake (Kcal/kg) was not an independent risk factor in this study, possibly because a proportion of patients who developed RFS were not considered at high risk for RFS at admission and did not restrict caloric intake.

  We have added the above information in the 6th paragraph of the Discussion section.

Q8. In Table 1 daily kcal should be better described as kcal/kg. Otherwise, it is rather meaningless.

Response: We appreciate the reviewer’s valuable suggestion and agree with it. We have redrawn calorie intake as kcal/kg in Table 1.

Q9. Line 198, consider changing “classified” for “identified”.

Response: We have replaced “identified” by “classified” in Line 211.

Q10. Line 264, consider changing “calories” by “energy supply”, for example

Response: We have substituted “energy supply” for “calories” in line 277.

Q11. Lines 294-297, 315-316, need rewriting.

As the findings of the authors is that malnutrition screening tools are poor predictors of RFS, they could speculate in the discussion about the pathophysiology of RFS and nutrient provision, independently of the nutrition status.

Response: Thanks for your constructive suggestion. We have reworked these contents in the Discussion section (311-325, and 342-344).

References:

  1. Da Silva, J.S.V.; Seres, D.S.; Sabino, K.; Adams, S.C.; Berdahl, G.J.; Citty, S.W.; Cober, M.P.; Evans, D.C.; Greaves, J.R.; Gura, K.M.; Michalski, A.; Plogsted, S.; Sacks, G.S.; Tucker, A.M.; Worthington, P.; Walker, R.N.; Ayers, P. ASPEN Consensus Recommendations for Refeeding Syndrome. NUTR CLIN PRACT 2020, 35, 178-195.
  2. Vahdat Shariatpanahi, Z.; Vahdat Shariatpanahi, M.; Shahbazi, E.; Shahbazi, S. Refeeding Syndrome and Its Related Factors in Critically Ill Coronavirus Disease 2019 Patients: A Prospective Cohort Study. Frontiers in Nutrition 2022, 9.
  3. Cederholm, T.; Jensen, G.L.; Correia, M.I.T.D.; Gonzalez, M.C.; Fukushima, R.; Higashiguchi, T.; Baptista, G.; Barazzoni, R.; Blaauw, R.; Coats, A.J.S.; Crivelli, A.N.; Evans, D.C.; Gramlich, L.; Fuchs Tarlovsky, V.; Keller, H.; Llido, L.; Malone, A.; Mogensen, K.M.; Morley, J.E.; Muscaritoli, M.; Nyulasi, I.; Pirlich, M.; Pisprasert, V.; Schueren, M.A.E.; Siltharm, S.; Singer, P.; Tappenden, K.; Velasco, N.; Waitzberg, D.; Yamwong, P.; Yu, J.; Van Gossum, A.; Compher, C. GLIM criteria for the diagnosis of malnutrition – A consensus report from the global clinical nutrition community. Journal of Cachexia, Sarcopenia and Muscle 2019, 10, 207-217.
  4. Doig, G.S.; Simpson, F.; Heighes, P.T.; Bellomo, R.; Chesher, D.; Caterson, I.D.; Reade, M.C.; Harrigan, P.W.J. Restricted versus continued standard caloric intake during the management of refeeding syndrome in critically ill adults: a randomised, parallel-group, multicentre, single-blind controlled trial. The Lancet Respiratory Medicine 2015, 3, 943-952.
  5. Friedli, N.; Stanga, Z.; Culkin, A.; Crook, M.; Laviano, A.; Sobotka, L.; Kressig, R.W.; Kondrup, J.; Mueller, B.; Schuetz, P. Management and prevention of refeeding syndrome in medical inpatients: An evidence-based and consensus-supported algorithm. NUTRITION 2018, 47, 13-20.
